# Exercise-Based Interventions Are Effective in the Management of Patients with Thumb Carpometacarpal Osteoarthritis: A Systematic Review and Meta-Analysis of Randomised Controlled Trials

**DOI:** 10.3390/healthcare12080823

**Published:** 2024-04-12

**Authors:** Stefanos Karanasios, Dimitra Mertyri, Fotis Karydis, George Gioftsos

**Affiliations:** 1Physiotherapy Department, School of Health and Care Sciences, University of West Attica, 122 43 Aigaleo, Greece; gioftsos@uniwa.gr; 2Hellenic OMT eDu, 116 31 Athens, Greece; dimitramertyri@gmail.com (D.M.); fokarid@gmail.com (F.K.)

**Keywords:** training, exercises, osteoarthritis, trapeziometacarpal, thumb, base

## Abstract

Exercise-based interventions are a common management strategy in patients with thumb carpometacarpal joint osteoarthritis (CMCJ OA); however, their exact effect on or the use of an optimal training programme for reducing pain and disability remains unclear. Our purpose was to evaluate the effectiveness of exercise-based interventions compared with other conservative interventions in patients with CMCJ OA. We performed a systematic review and meta-analysis based on the Preferred Reporting Items for Systematic Reviews and Meta-Analyses (PRISMA) guidelines. Fourteen randomised clinical trials with 1280 patients were finally included. Exercise-based interventions present statistically and clinically better outcomes in reducing pain intensity (mean difference [MD]: −21.91; 95% confidence interval [CI]: −36.59, −7.24; *p* = 0.003) and wrist disability (MD: −8.1, 95% CI: −4.6, −11.5; *p* = 0.02) compared with no treatment at short-term follow-up. Proprioceptive exercises have statistically and clinically better outcomes compared with standard care only in pain intensity at very short-term (standardised mean difference [SMD]: −0.76; 95% CI: −1.30, −0.21; *p* = 0.007) and short-term (SMD: −0.93; 95% CI: −1.86, −0.01; *p* = 0.049) follow-up and statistically better results in wrist disability at very short-term (SMD: −0.94; 95% CI: −1.68, −0.21; *p* = 0.01) follow-up. No differences were found between the comparators at mid- and long-term follow-up. Low to moderate certainty of evidence suggests that exercise-based interventions can provide clinically better outcomes compared with no treatment in patients with thumb CMCJ OA, at least in the short term.

## 1. Introduction

Thumb carpometacarpal joint (CMCJ) osteoarthritis (OA) is a common degenerative disease of the hand causing severe pain, stiffness, weakness and increased functional loss [1]. The prevalence of the condition can reach up to 7% of men and 15% of women over 50 years of age [1,2]. Several risk factors have been identified for the development of the thumb CMCJ OA including age, jobs involving repetitive use of the thumb and post-menopausal period [2]. The manifestation of the thumb CMCJ OA is based on the gradual trapeziometacarpal joint degeneration including deterioration of the articular surfaces, osteophytes formation and ligamentous laxity [1]. As a result, in the chronic stages of the condition, patients present decreased neuromuscular control of the joint, subluxation and adduction contracture of the thumb [3]. 

Although surgical interventions can be very effective in providing relief in patients with thumb CMCJ OA, they are considered the last treatment option due to the underlying risk of complications [4]. Hence, multimodal conservative interventions are commonly proposed for first-line management of patients with CMCJ OA [4]. The non-surgical approach may include non-steroid anti-inflammatory drugs, education, therapeutic exercises, injections (corticosteroid or hyaluronic), physiotherapy modalities, manual therapy and orthotic devices [4]. Previously published systematic reviews supported that therapeutic exercises along with manual therapy and orthotic interventions can be effective in improving pain and function at short-term follow-up in patients with CMCJ OA [5,6]; however, their recommendations should be considered cautiously due to the low methodological quality and the limited numbers of trials included. In the same line, Spaans et al. (2020) in their systematic review advocated that ‘hand therapy’ and intra-articular injections are effective interventions in the management of CMCJ OA, at least in the short term [7]; nevertheless, the effect of hand exercise programmes was poorly reported. Also, several trials evaluating the effectiveness of different training programmes in CMCJ OA have been published recently; however, they have not been considered in a systematic review yet. Therefore, a research synthesis on the effectiveness of exercises compared with other interventions seems necessary.

Therefore, the aim of our systematic review was to examine the effectiveness of exercises either alone or as an additive intervention to a multimodal treatment regime compared with other conservative interventions in the management of thumb CMCJ OA. We planned to include subgroup analyses comparing exercises with (i) control, (ii) other interventions or other types of exercises at different follow-up periods were applicable. The certainty of evidence was assessed using published recommendations [8].

## 2. Materials and Methods

Our review was conducted following the Preferred Reporting Items for Systematic Reviews and Meta-Analyses (PRISMA) guidelines [9]. Before commencing the study, its protocol was registered in the Prospective Register of Systematic Review (PROSPERO) (CRD42023461505).

### 2.1. Eligibility Criteria

#### 2.1.1. Study Design

Only randomised controlled trials (RCTs) were considered eligible for the present review. Trials should evaluate the effectiveness of exercises used either alone or as an additive intervention to a multimodal therapy programme compared with other conservative interventions. We applied no language restrictions in the search strategy and study selection.

#### 2.1.2. Population

The population included only adult patients of both sexes with thumb CMCJ OA due to degeneration, overuse or injury. The diagnosis could be confirmed clinically, radiologically or both. Studies including patients managed surgically or diagnosed with rheumatic diseases, peripheral or central neurological deficits, De Quervain tenosynovitis or Dupuytren’s syndrome were excluded from the present review.

#### 2.1.3. Interventions

Interventions included any type of exercise, such as stretching, isometric, isotonic, concentric or eccentric contractions (or a combination thereof). Exercises could include equipment such as elastic bands, elastic bars, dumbbells or squeeze balls. The exercise programmes could be used alone or in addition to other interventions. Also, they could be supervised or prescribed as a home exercise programme. 

#### 2.1.4. Comparisons

We included studies that compared exercise-based interventions with any type of non-surgical treatment, control, placebo or sham interventions. Also, we included studies including comparisons between different types of training programmes (i.e., strengthening, proprioceptive, nerve gliding, supervised or unsupervised, etc.).

#### 2.1.5. Outcomes

The primary outcomes of our review were the following: (i) pain intensity using visual analogue scale (VAS) score or numeric pain rating scale (NPRS), (ii) disability using validated patient-reported outcome measures such as the full or quick version of the Disability of the Arm, Shoulder and Hand (DASH) questionnaire, the Upper Limb Functional Index, and the Functional Index of Hand OA (FIHOA)/Dreiser Index. As secondary outcome measures, we considered (i) grip or pinch strength, (ii) range of motion, (iii) pressure pain thresholds, and (iv) joint position error. Outcomes were grouped by the timing of follow-up: (i) very short term (≤2 months), (ii) short term (>2 months ≤3 months), (iii) mid term (>3 to <12 months) and, (iv) long term (≥12 months) [10].

### 2.2. Search Strategy

The PubMed, CINAHL, EMBASE, PEDro, ScienceDirect, Cochrane Library, and grey literature databases and clinical trial registries were systematically searched from inception to November 2023. The full search strategy is presented in online Appendix A.

### 2.3. Data Selection

Two reviewers (DM and FK) independently evaluated the search results against eligibility criteria in two stages [9]. To resolve any discrepancies, a consensus process was followed including the following standards: if the two reviewers have reached different judgements, the conflict was resolved by a discussion between them. If there was no consensus, a referral to a third reviewer (SK) was made for the final decision. All disagreements were recorded in a separate record. 

### 2.4. Data Extraction

The same independent researchers extracted the data from included articles using standardized forms (citation details, sample size, treatment groups, outcomes, follow-up and main results). To resolve disagreements in this stage, a third reviewer (SK) was used following a consensus approach similar to the data selection stage.

### 2.5. Risk of Bias

The methodological quality of the studies was assessed by two reviewers (DM and FK) independently using the PEDro scale. The current tool is a valid and reliable tool for evaluating the quality of a study and includes a score from 0 to 10 [11,12]. For a PEDro score ≤ 4, the methodological quality was considered ‘poor’, for scores of 5 or 6 ‘moderate’ and for scores ≥ 7 ‘high’ [11,13]. Any disagreements were discussed in a consensus meeting including a third reviewer (GG).

### 2.6. Data Analysis, Synthesis and Summary of Findings

To facilitate data synthesis, all continuous data were converted into 100-point scales. To evaluate the treatment effect, we used the mean differences (MDs) with 95% CIs when the outcome was measured in the same unit or the standardised mean differences (SMDs) with 95% CIs for different units [14]. Considering the substantial clinical and methodological heterogeneity across the eligible trials, we used a random-effect model to pool the outcomes of the studies. To assess the risk of publication bias, we aimed to examine the asymmetry of funnel plots if a comparison included at least ten trials [8]. We used the Cochrane Review Manager Software V.5.4 (The Cochrane Collaboration) to estimate the meta-analysis results.

The I^2^ statistic that aims to indicate the proportion of the non-random variation in the treatment effect was used to estimate the heterogeneity among the studies included in the meta-analysis [8]. I^2^ values ≥ 0.75 were considered of high heterogeneity [8]. We performed subgroup analyses between exercise-based interventions and (i) no intervention (sham/placebo or control), (ii) other types of intervention (such as modalities and corticosteroid injections), and (iii) other types of exercises. Sensitivity analyses (post hoc registration) were conducted in comparisons including studies with ‘low’ or ‘moderate quality’ (PEDro score < 7) or unexpectedly large treatment effect sizes. Statistical significance (*p*) was set at <0.05.

We used Grading of Recommendations, Assessment, Development, and Evaluations (GRADE) to summarize the evidence. GRADE is a transparent framework that assesses certainty of evidence based on the following items: study design, high risk of bias (low or moderate quality in PEDro score in >75% of the eligible studies), inconsistency (substantial heterogeneity on the point estimates, statistical heterogeneity and I^2^ > 50%), indirect evidence, imprecision (the sample did not reflect inclusion criteria of the review, CIs limit crossed the effect size of 0.5) and publication bias [8]. Initially, the evidence was rated for each comparison as ‘high certainty’ and was downgraded for any of the previous reasons [10]. In comparison, including one trial, the evidence was graded as low certainty, and if this study was evaluated with low-quality evidence, it was graded as very low certainty [13,15].

To evaluate the minimal clinically important differences (MCIDs) we have used the effect size value [16]. Based on Cohen’s d benchmark, we considered effect sizes below 0.2 as low, 0.5 as medium and 0.8 or higher as high [17]. We calculated MCIDs by dividing the mean difference in scores by the standard deviation of baseline scores and values of 0.5 were considered clinically significant [18].

## 3. Results

### 3.1. Studies Selection

The main research identified 2624 records. After the removal of duplicates, 1386 records were considered relevant. Subsequent screening of titles and abstracts resulted in 29 eligible publications for full-text assessment. Fifteen studies did not satisfy the inclusion criteria and fourteen RCTs were finally included [19,20,21,22,23,24,25,26,27,28,29,30,31,32]. The flow chart of the study selection is presented in Figure 1.

### 3.2. Participants

The total number of participants was 1280 with a mean age of 62.2 years. The majority of the participants were female (79.7%). The sample sizes ranged from 12 to 204 participants. The diagnosis of thumb CMCJ OA was confirmed using the radiographic Eaton-Littler Classification or the Modified Kellgren–Lawrence grade scale (grades 0–4, 0 = no OA) [33,34]. Three studies included patients with CMCJ OA grade 1 to 4 [26,29,32], four studies with grade 1 to 3 [20,22,23,27], three studies with grade 3 or 4 [23,27,28], two studies with grade 2 to 4 [30,31], two studies with grade 1 or 2 [21,28] and one study with grade 3 [24]. A detailed description of the study characteristics and main results are shown in Table 1. 

### 3.3. Description of the Studies

Exercise as an intervention was used alone or in combination with a multimodal physiotherapy programme [20,21,22,26,27,28,29,30] and orthotic devices [20,22,23,25,26,27,32]. Eight trials evaluated the use of exercises against a multimodal therapeutic programme [20,21,22,23,26,27,30,31], another exercise programme [19,24], no treatment [29,32], joint protection alone [25] or corticosteroid injections [28]. Exercise-based therapy programmes were used for between 2 and 12 weeks. Nine of the eligible trials evaluated the use of proprioceptive exercise programmes [19,20,21,22,23,24,25,26,27]. Three of the eligible trials evaluated the use of mobility and strengthening exercises while two trials focused on the use of neurodynamic exercises (Table 1).

### 3.4. Risk of Bias within Studies

The quality assessment according to the PEDro criteria showed the quality of six RCTs as ‘high’, five RCTs as ‘moderate’ and three RCTs as ‘low quality’ (Table 2). Half of the eligible studies lacked a concealed allocation and none of them ensured the blinding of the participants and therapists. Blinding the outcome assessors was not ensured in more than one-third of the eligible trials. Also, a substantial proportion of the trials (>35%) reported a significant drop-out rate (>15%) and did not include an intention-to-treat analysis. All outcomes were rated from very low to moderate certainty of evidence.

### 3.5. Meta-Analysis Results

#### 3.5.1. Exercise Compared with No Treatment or Corticosteroid Injections

Two RCTs [29,30] made an indirect comparison between exercises in addition to a multimodal therapy programme and a control intervention for thumb CMCJ OA. The maximum duration of follow-up was three months and the mean age of the participants (*n* = 240) was 67.8 years. There was a statistically and clinically significant difference in favour of exercises with manual therapy and orthoses compared with control interventions in pain intensity at short-term follow-up (MD: −21.91; 95% CI: −36.59, −7.24; *p* = 0.003, d = 0.93) (Appendix A). One RCT [29] evaluated wrist disability using the QuickDASH score and suggested statistically and clinically better outcomes in favour of exercises at short-term follow-up (MD: −8.1, 95% CI: −4.6, −11.5; *p* = 0.02, d = 0.51) (Appendix A). Two studies evaluated pinch strength suggesting no difference between the comparators at short-term follow-up (Appendix A) [29,30]. A forest plot for the effectiveness of exercises in addition to a multimodal treatment programme compared with no treatment in pain intensity is shown in Figure 2.

One RCT evaluated the use of exercises in addition to a multimodal therapy programme against corticosteroid injections in pain intensity and disability [28]. Based on very low-quality evidence, there was a statistically and clinically significant difference in disability scores in favour of corticosteroid injections at short-term follow-up (MD: 3.6; 95% CI: 2.94, 4.26; *p* < 0.05, d = 0.54); however, no differences were found between the comparators in all outcomes on the short-, mid- and long-term occasions (Table 2). One RCT [31] compared a radial nerve mobilisation exercise and a placebo intervention, suggesting significant differences in pressure pain thresholds at the CMCJ in favour of the experimental group at very short-term follow-up (MD: 1.58, 95% CI: 0.73, 2.42; *p* < 0.001). 

#### 3.5.2. Proprioceptive Exercises Compared with Standard Treatment

Nine RCTs [20,21,22,23,24,25,27,30,31] compared proprioceptive exercises with standard therapy for thumb CMCJ OA. The outcomes were assessed at very short-, short-, mid- and long-term follow-up times. In total, 750 participants were included among the eligible studies with a mean age range between 58 and 67.1 years (Table 1).

Two out of nine eligible studies did not report the mean values of the study groups and subsequently, were excluded from quantitative synthesis [24,25]. We found a statistically and clinically significant difference in favour of proprioceptive exercises compared with standard treatment in mean change in pain intensity at very short- (SMD: −0.76, 95% CI: −1.30, −0.21; *p* = 0.007, d = 0.52) and short-term (SMD: −0.93, 95% CI: −1.86, −0.00; *p* = 0.05, d = 0.54) follow-up (Appendix A). There was no difference between proprioceptive exercises and standard treatment in pain intensity at the mid- (SMD: 0.26, 95% CI: −0.04, 0.55; *p* = 0.09) and long-term (SMD: −0.05, 95% CI: −0.38, 0.49; *p* = 0.80) follow-up times. Although a statistically significant difference was found in favour of proprioceptive exercises in disability scores in the very short-term (SMD: −0.94, 95% CI: −1.68, −0.21; *p* = 0.01), no differences were observed at short- (SMD: −0.81, 95% CI: −1.84, 0.23; *p*= 0.13), mid- (SMD: −0.14, 95% CI: −0.25, 0.52; *p* = 0.49) and long-term follow-up (SMD: −0.03, 95% CI: −0.47, 0.41; *p* = 0.89) occasions (Appendix A, Figure 3 and Figure 4).

## 4. Discussion

### 4.1. Main Findings and Comparison with Other Reviews

Based on our knowledge, this is the first systematic review and meta-analysis evaluating the effectiveness of exercise-based therapy programmes compared with other conservative interventions in the management of patients with thumb CMCJ OA. We analysed 14 RCTs including 1280 patients with thumb CMCJ OA with a mean age of 62.2 years. Most of the eligible studies presented high (6) or moderate (5) methodological quality. Subgroup analyses presented substantial statistical heterogeneity, inconsistency and indirectness of interventions; therefore, the results were rated between very low and moderate certainty of evidence.

Our findings suggest moderate certainty evidence of statistically and clinically significant benefits for an exercise component compared with control interventions in pain intensity and wrist disability at short-term follow-up. Also, based on low certainty evidence, interventions with proprioceptive exercises outperform standard therapy in mean change in pain intensity at very short- and short-term follow-up and in wrist disability (at very short-term follow-up). However, there are no differences among the different types of exercises in wrist disability in the short term or in pain and disability on mid- and long-term follow-up occasions.

Several systematic reviews and meta-analyses advocated the effectiveness of a range of conservative interventions in patients with thumb CMCJ OA such as manual therapy [36], splints [37,38], injections [39] and multimodal physiotherapy interventions [5,40]. Considering the objectives of the present review, a direct overall comparison between our results with the systematic reviews that do not contain an active component is difficult. Nevertheless, our findings are in agreement with two previous systematic reviews and meta-analyses that suggested low and very low certainty of evidence for the superiority of unimodal or multimodal physiotherapy interventions including exercises in thumb CMCJ OA in pain and function in the short term [5,41]. However, the results of these reviews should be interpreted with caution because both studies included only three RCTs that evaluated the use of exercises in thumb CMCJ OA.

Also, our findings regarding the equivocal effectiveness of an exercise-based intervention and corticosteroid injections in pain and function at mid- and long-term follow-up were in agreement with a previous meta-analysis by Riley et al. (2018) suggesting similar benefits between injection- and non-injection-based interventions in patients with thumb CMCJ OA [39]. However, the limited number of eligible RCTs and the low quality of evidence require careful consideration of the current results.

### 4.2. Exercise-Based Interventions Have Better Results Than No Treatment

We found a short-term statistically and clinically significant benefit in pain intensity and function for the use of exercises in addition to multimodal therapy programmes compared with control interventions. In the same line, one eligible trial including 180 patients with thumb CMCJ OA suggested that a three-month multimodal therapy with exercises can delay and reduce the need for surgery at a two-year follow-up [32]. Overall, our findings were similar to a Cochrane review supporting the thesis that exercise-based interventions consistently contribute to improvement regarding pain and hand function in patients with hand osteoarthritis [42].

It is well documented that different types of exercises, such as aerobic, neuromuscular, proprioceptive and resistance training, can act as pain-modulation interventions in patients with OA [43]. The most significant underlying mechanism for this phenomenon is considered the activation of descending inhibitory pathways resulting in widespread hypoalgesia even after a single bout of exercise [31,43]. Considering that significant widespread sensitization has been identified in patients with CMCJ OA, exercise-induced hypoalgesia can be a valuable option to improve the treatment outcomes in patients with the condition [19]. Also, exercise-based interventions have been reported to relieve several pathological mechanisms of OA [43]. Some of these mechanisms include the degradation of the extracellular matrix, apoptosis, inflammation and other cellular changes [43,44,45,46]. 

Despite the overall positive clinical outcomes for the use of exercise-based interventions compared to no treatment in pain intensity and disability scores, there was no difference in pinch strength. This discrepancy may be attributed to the fact that exercise programmes did not include exercises focusing on pinch strength, avoiding the risk of causing symptoms to worsen due to hypermobility and subluxation of the thumb [29,30]. According to our analysis, there is substantial clinical heterogeneity regarding the type (multimodal therapy programme) and time course of the intervention periods (four to 12 weeks) proposed. Hence, despite the evidence that exercise-based multimodal treatments can be effective in key outcomes such as pain and function in the short term, further knowledge regarding the parameters of a long-term optimal treatment regime in multiple parameters is required. 

### 4.3. Proprioceptive Exercises Are More Beneficial Than Standard Treatment Only in the Short Term

In terms of the most effective type of exercise during the management of thumb CMCJ OA, proprioceptive exercises seem to perform better compared with standard treatment in pain intensity and disability scores only at very short- and short-term follow-up. Although we found no differences between the comparators at mid- and long-term follow-up, the clinical interpretation of the current finding should be considered with caution due to the limited number of eligible RCTs on the mid- (2) and long-term follow-up (1) occasions.

The presence of pain in patients with thumb CMCJ OA significantly affects functional activities and is usually the most important reason for patients seeking rehabilitative services [6,37]. Therefore, interventions including manual therapy or orthotic interventions are commonly used to decrease pain in the current patient group [20,36]. However, longstanding OA involves complex pathophysiological mechanisms resulting in altered pain transmission and several changes within the joint afferent neurons and the central nervous system [47,48,49]. Thus, there is increased recognition that the management of OA interventions should include exercises and education regarding pain to effectively decrease pain intensity, increase self-efficacy and improve social functioning [47]. Contemporary evidence suggests that the presence of osteoarthritis is associated with a proprioceptive deficit in thumb CMCJ OA [30,50]. A logical assumption is that proprioceptive exercises may enhance joint stability, improve osteoarticular coordination, increase functional performance in daily activities and result in pain reduction. Nevertheless, based on the present findings, the use of proprioceptive exercises does not present superior clinical benefits compared to standard care in thumb CMCJ OA at the long-term follow-up.

### 4.4. Limitations and Future Research

Our systematic review and meta-analysis should be viewed in the light of some limitations. First, we were not able to conduct a subgroup analysis regarding several confounders that may have influenced the outcomes, such as the duration of symptoms or the participants’ grade of thumb CMCJ OA. Also, although we graded the evidence as low certainty when one trial was available and downgraded the level of evidence if the trial was of low quality, this method lacks validation [15]. Our quantitative synthesis included both direct and indirect comparisons that might underlie a high risk of imprecision estimates [51]. The low number of studies in most comparisons limited our ability to generate funnel plots to assess for publication bias.

We suggest that future research should focus on the evaluation of the effectiveness of exercise-based interventions using confounders such as the severity or duration of symptoms, the grade of OA, and age. Also, further research on the effectiveness of exercise programmes compared with other long-term interventions is necessary.

## 5. Conclusions

Based on the available data, exercise-based interventions provide statistically and clinically significant benefits in pain reduction and wrist disability compared with no treatment in patients with thumb CMCJ OA in the short term. Also, the use of proprioceptive exercises provides statistically and clinically better outcomes compared to other types of exercises only in pain intensity in the short-term without further differences in the outcomes at mid- and long-term follow-up times. Future research evaluating the effect of exercise-based interventions in different patient subgroups is necessary.

## Figures and Tables

**Figure 1 healthcare-12-00823-f001:**
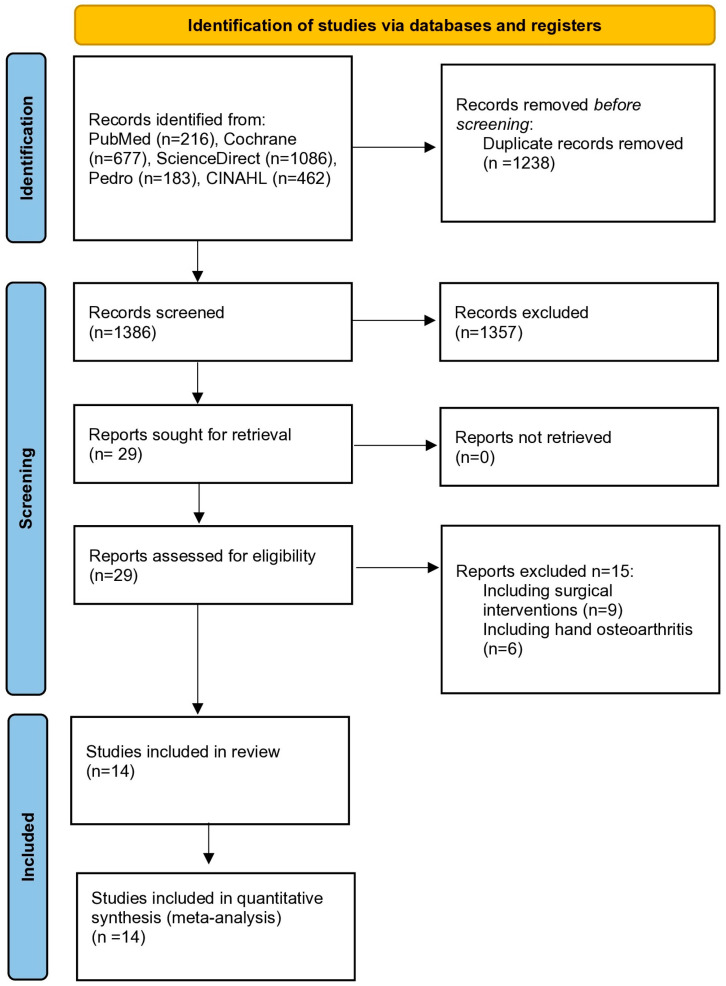
PRISMA study selection flow chart. PRISMA: Preferred Reporting Items for Systematic Reviews and Meta-Analyses.

**Figure 2 healthcare-12-00823-f002:**
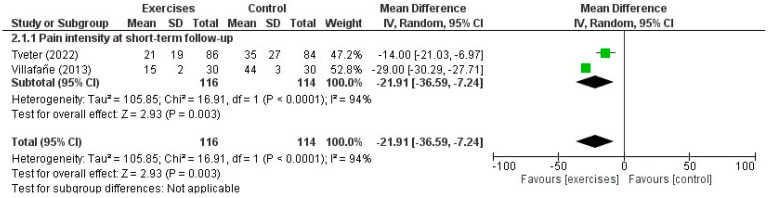
Forest plot for the effectiveness of the use of exercises in addition to a multimodal therapy programme compared with a control group in patients with thumb carpometacarpal osteoarthritis [29,35].

**Figure 3 healthcare-12-00823-f003:**
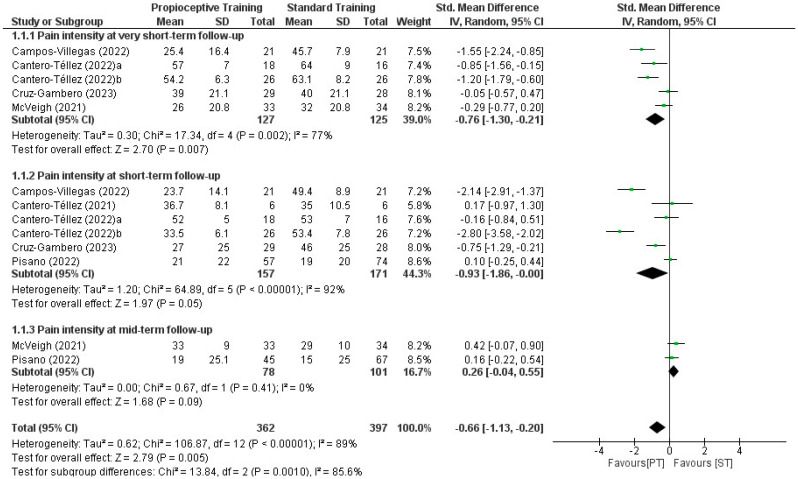
Forest plot for the effectiveness of the use of proprioceptive training compared with standard treatment in pain intensity in patients with thumb carpometacarpal osteoarthritis [19,20,21,22,23,26,27].

**Figure 4 healthcare-12-00823-f004:**
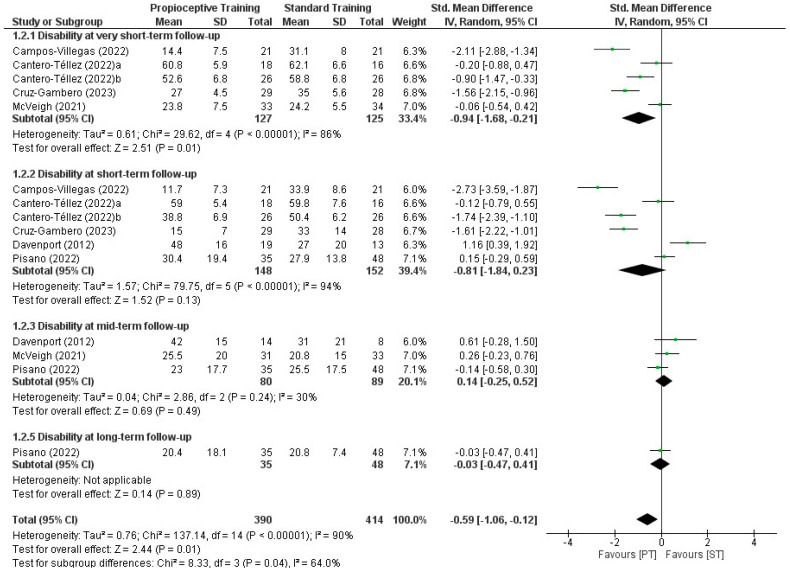
Forest plot for the effectiveness of the use of proprioceptive training compared with standard treatment in wrist disability in patients with thumb carpometacarpal osteoarthritis [19,20,22,23,24,26,27].

**Table 1 healthcare-12-00823-t001:** Included studies, demographics and results.

Study (Year)	Total Sample Size *n* * and Age **	Interventions	Length of Follow-Up	Outcome Measures	Results
Villafañe, (2013) [30]	60 total EG: *n* = 30 (82 ± 2 y); (0 lost to 2 months follow-up) CG: *n* = 30 (83 ± 1 y); (0 lost to 2 months follow-up)	EG: grade 3 posterior/anterior glide with distraction technique to the 1st CMC, 3 × 3 min, with 1 min rest periods. Passive nerve slider for the medial nerve, 2 sets × 5 min with 1 min rest period, 6 active ROM exercises, 3 exercises for grip and pinch strength with a nonlatex polymer ball (10 reps for the first 4 sessions, 12 reps for the next 2 sessions, 15 reps for the next 2 sessions and 20 for the last 4 sessions, if they were able), (12 sessions over 4 weeks) CG: Inactive doses of pulsed US to the hypothenar area, 10 min/session, 12 sessions over 4 weeks.	immediately after the intervention, 4 and 12 weeks	Pain intensity (VAS) PPTs Grip & Pinch strength	EG had significantly greater reduction in pain intensity. No differences in PPTs, grip and pinch strength.
Villafañe (2013) [35]	60 total EG: *n* = 30 (81 ± 7 y); 0 lost to follow-up CG: *n* = 30 (82 ± 7 y); 0 lost to follow-up	EG: Passive neurodynamic radial nerve slider technique over the symptomatic hand, 6 sessions over 4 weeks, 3 × 3 min, with 1 min rest periods. CG: Inactive doses of pulsed US to the hypothenar area, 10 min/session, 6 sessions over 4 weeks	immediately after the intervention, 4, 8 and 12 weeks	PPTs	Significant changes in favour of EG in PPTs.
Tveter (2022) [29]	180 total EG: *n* = 90 (62.8 ± 7.5 y); 4 lost to follow up CG: *n* = 90 (63.3 ± 7.8); 6 lost to follow up	EG: patient education, exercises at home (joint mobility, grip strength and stability of the wrist and the fingers), orthoses (daytime and night time) for the CMC 1 joint, 5 assistive devices. 3 times/week for 12 weeks CG: no treatment, oral and written information about OA	92–106 days	Pain (NRPS) Grip & Pinch strength ROM MAP-Hand and QuickDASH	Significantly better results in favour of EG group in pain intensity at rest, pain following grip strength, grip strength, MAP-Hand and QuickDASH.
Rocchi (2017) [28]	50 total (62 ± 6 y) Physiotherapy: *n* = 25; 2 lost to follow-up Corticosteroid inj: *n* = 25; 1 lost to follow-up	Physiotherapy: heat, passive and active mobilization of the TMC joint, massage therapy, stretching of the first web span, TMC splinting (4 weeks, full time) 30–40 min/session, 10 sessions over 2 weeks CI: methylprednisolone acetate (40 mg/1 mL) and lidocaine (10 mg), TMC splinting (4 weeks, full time)	8, 26 and 52 weeks	Pain and Restriction scale DASH Overall treatment satisfaction Pinch strength	Corticosteroid injections showed significantly better results in pain, function and strength, at 8 weeks. No between-group difference at 1-year follow-up
Pisano (2022) [27]	190 total Standard Therapy + stabilization + HEP: *n* = 96 (61 ± 9.6 y); 48 lost to follow up Standard Therapy: *n* = 94 (60 ± 8.8 y); 59 lost to follow up	Standard Therapy + stabilization+ HEP: Standard care plus a stretching and/or stabilizing and/or strengthening program for the thumbs Standard Therapy: heat modalities, joint protection education, adaptive equipment training and orthosis.	8–12, 26 and 52 weeks	Pain intensity QuickDASH AROM/PROM PSFS Thumb, grip and pinch strength	Non-significant differences between groups in all outcomes at all follow-up times.
McVeigh (2021) [26]	84 total Standard Therapy + HEP: *n* = 42 (66.5); 11 lost to follow up Standard Therapy: *n* = 42 (63.9 y); 9 lost to follow up	Standard Therapy + HEP: instructions, adaptive equipment, orthosis, home exercise programme, adductor stretching (90 s, 3 reps daily), isometric contractions with pinch (5–10 s, 10 reps × 3 times daily), isometric first dorsal interosseous strengthening (5–10 s, 10 reps × 3 times daily) Standard Therapy: instructions, adaptive equipment, orthosis	6 and 26 weeks	Pain intensity QuickDASH ROM Kapandji opposite scale Pain-free grip strength Pinch strength	Non-statistically significant between-group differences in pain intensity and quickDASH.
Cruz-Gambero (2023) [23]	83 total (60 ± 7 y) EG: *n* = 41; 13 lost to follow-up CG: *n* = 42; 13 lost to follow-up	EG: orthosis, exercises and HEP including specific proprioceptive exercises CG: orthosis and exercises	4 and 12 weeks	Pain intensity JPS FST COPM	Significantly better results in favour of EG group in pain intensity and COMP at 12 weeks follow-up. Non-significant differences between groups in JPS and FST.
Gravas (2019) [32]	180 total Occupational therapy: *n* = 90 (62.8 ± 7.5 y); 6 lost to follow-up CG: *n* = 90 (63.3 ± 7.8 y); 7 lost to follow-up	Occupational therapy: Oral and written education, assistive devices, day and night CMCJ orthoses and a hand exercise programme. Anti-inflammatory and pain relief medication usage was allowed. CG: Oral and written education. Anti-inflammatory and pain relief medication usage was allowed.	12 weeks, 18 months and 2 years	The proportion of patients in each group that received surgery after 2 years	Non-statistically significant between-group differences in the proportion of patients that received surgery after 2 years.
Davenport (2012) [24]	38 total Stabilizing Exercises: *n* = 17 (58 ± 11 y); 9 lost to follow-up Strengthening Exercises: *n* = 21 (61 ± 10 y); 7 lost to follow-up	Stabilizing Exercises: CMC joint stabilizing exercises 3–4 times daily, 3 sets of 10 reps including passive and active exercises progressing with resistance using an elastic band and eventually pinching and turning or twisting functional exercises. Strengthening Exercises: General strengthening exercises, 3–4 times daily, 3 sets of 10 reps including passive and active exercises progressing with resistance using a peg or a sponge, and eventually pinching and turning or twisting functional exercises	12 and 26 weeks	Pain intensity at rest and during pinch DASH Pinch strength	No statistically significant difference between groups in all outcomes at both follow-up times.
Deveza (2021) [25]	204 total EG: *n* = 102 (64.72 ± 12.02 y; 6 lost to follow-up) CG: *n* = 102 (65.20 ± 8.46 y); 3 lost to follow-up	EG: Education and joint protection, splint use, thumb base joint exercises with a stress ball, chopsticks or the gravity itself (3 sessions/week), NSAID 3 times/day CG: Education and joint protection	2, 6 and 12 weeks	Pain intensity FIHOA Grip strength Tip-pinch strength PGA	Patients with lower radial subluxation had better results in pain intensity at 6 weeks follow-up. No--statistically significant differences were found in the rest of the outcomes.
Cantero-Téllez (2021) [21]	12 total EG: 6 (67.17 y) CG: 6 (65.33 y)	EG: 4-week exercise program including active and/or resistive exercises, night splinting, self-passive traction of the thumb, self-massage, a HEP plus a 3-phase training programme with proprioceptive exercises CG: 4-week exercise program including active and/or resistive exercises, night splinting, self-passive traction of the thumb, self-massage, and a HEP	3 months follow up	Pain intensity Pinch strength JPS error	Not statistically significant difference at 3 months in pain intensity and pinch strength between groups. Statistically significant differences in JPS error in favour of EG
Cantero-Téllez (2022) [22]	45 total EG: *n* = 22 (63 ± 7 y) CG: *n* = 23 (62 ± 7 y)	EG: 4-week exercise program including active and/or resistive exercises, self-passive traction of the thumb, self-massage, a HEP plus proprioceptive exercises CG: 4-week exercise program including active and/or resistive exercises, self-passive traction of the thumb, self-massage and HEP	4 and 12 weeks	Pain intensity ADL QuickDACH COMP JPS error	Statistically significant differences between groups in pain intensity at 4 weeks and in JPS error at 12 weeks in favour of the EG. Non-significant differences in pain intensity at 12 weeks, ADL, quickDASH and COMP at 4- and 12-week follow-up times.
Cantero-Téllez (2022) [20]	52 total EG: *n* = 26 (63.5 ± 6.6 y) CG: *n* = 26 (62.7 ± 7.9 y)	EG: 4-week exercise program including active and/or resistive exercises, night splinting, self-passive traction of the thumb, self-massage, a HEP plus a 3-phase training programme with proprioceptive exercises CG: 4-week exercise program including active and/or resistive exercises, night splinting, self-passive traction of the thumb, self-massage, and a HEP	4 weeks 12 weeks	Pain intensity Quick DASH COMP JPS error	Statistically significant difference at 4 weeks and at 12 weeks between groups in favour of the EG in NRS Not statistically significant difference between groups at 4 weeks and 12 weeks in Quick-DASH, COMP and JPS
Campos-Villegas (2022) [19]	42 total Proprioceptive neuro-facilitation: *n* = 21 (59.14 ± 8.05 y) Strength training: *n* = 21 (61.04 ± 6.11 y)	PNFG: 4-week strength training, including warm-up exercises focusing on joint mobility and strength exercises plus rhythmic stabilization for the thumb, 3 times/week Strength training: 4-week strength training including warm-up exercises focusing on joint mobility and strength exercises, 3 times/week	4 and 8 weeks	Pain intensity DASH Grip pinch Palmar pinch Tip pinch Key pinch	Statistically better results in favour of Proprioceptive neuro-facilitation in pain intensity, DASH, grip pinch, palmar pinch, tip pinch and key pinch at both follow-up time points

* total number of participants; ** mean age of the participants.

**Table 2 healthcare-12-00823-t002:** Methodological quality assessment using the PEDro scale.

	1	2	3	4	5	6	7	8	9	10	11	Total Score
Villafañe (2013) [30]	−	+	−	+	−	−	+	+	+	+	+	7/10
Villafañe (2013) [35]	+	+	+	+	−	−	+	+	+	+	+	8/10
Tveter (2022) [29]	+	+	+	+	−	−	−	+	+	+	+	7/10
Rocchi (2017) [28]	+	−	−	−	−	−	−	+	+	+	+	4/10
Pisano (2022) [27]	+	+	−	+	−	−	−	−	−	+	+	4/10
McVeigh (2021) [26]	−	+	+	+	−	−	−	−	+	+	+	6/10
Cruz-Gambero (2023) [23]	+	+	−	+	−	−	−	−	−	+	+	4/10
Gravas (2019) [32]	+	+	+	+	−	−	+	+	+	+	+	8/10
Davenport (2012) [24]	+	+	+	+	−	−	−	−	−	+	+	5/10
Deveza (2021) [25]	+	+	+	+	−	−	+	+	+	+	+	8/10
Cantero-Téllez (2021) [21]	+	+	−	+	−	−	+	+	−	+	+	6/10
Cantero-Téllez (2022) [22]	+	+	−	+	−	−	+	+	−	+	+	6/10
Cantero-Téllez (2022) [20]	+	+	−	+	−	−	+	−	+	+	+	6/10
Campos-Villegas (2022) [19]	+	+	+	+	−	−	+	+	+	+	+	8/10

1. Eligibility criteria were specified. 2. Subjects were randomly allocated to groups; in a crossover study, subjects were randomly allocated to an order in which treatments were received. 3. Allocation was concealed. 4. The groups were similar at baseline regarding the most important prognostic indicators. 5. There was blinding of all subjects. 6. There was blinding of all therapists who administered the therapy. 7. There was blinding of all assessors who measured at least one key outcome. 8. Measures of at least one key outcome were obtained from more than 85% of the subjects initially allocated to groups. 9. All subjects for whom outcome measures were available received the treatment or control condition as allocated or, where this was not the case, data for at least one key outcome were analysed by “intention to treat”. 10. The results of between-group statistical comparisons were reported for at least one key outcome. 11. The study provides both point measures and measures of variability for at least one key outcome. Note: The first item relates to external validity and the remaining ten items are used to calculate the total score, which ranges from 0 to 10. + Yes
− No.

## Data Availability

Not applicable.

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
