# Peer review of "Exercise-Based Interventions Are Effective in the Management of Patients with Thumb Carpometacarpal Osteoarthritis: A Systematic Review and Meta-Analysis of Randomised Controlled Trials"

_healthcare, 2024, doi:10.3390/healthcare12080823_

Round 1
Reviewer 1 Report
Comments and Suggestions for Authors
We are delighted to be reviewing a paper on a very interesting topic. There is nothing I would like to point out regarding the content, but the figures and tables, including the flow chart, seem to need editing to fit the format of the journal or for readability.
Comments:
1) In the introduction, it was said that CMCJ OA is a disease that commonly occurs after the age of 40, but the condition for experiment participants was defined as being over 18 years old. Is there a special reason?
2) It is believed that comparative analysis of the effects of interventions will be possible only when the cause of OA is the same. Please also provide criteria for the cause of OA in the selection or exclusion criteria.
3) In Comparison, there is an expression 'other type of exercise', but for the reader's clear understanding, it is necessary to know the specific meaning.
4) We believe that accurate standards are important for selecting and extracting data in review papers. Discrepancies in data selection and extraction are described, and it is stated that a consensus process was conducted including a third reviewer. Please describe by what standards the consensus process, including a third reviewer, was conducted.
Author Response
We would like to thank the reviewer for the comments. All comments were considered and changes were made in the revised document according to your recommendations.
|
Comments |
Responses |
|
Reviewer We are delighted to be reviewing a paper on a very interesting topic. There is nothing I would like to point out regarding the content, but the figures and tables, including the flow chart, seem to need editing to fit the format of the journal or for readability. |
Thank you for your kind feedback. The figures were uploaded in a higher resolution and tables were improved to improve readability. |
|
Comments to the Author: In the introduction, it was said that CMCJ OA is a disease that commonly occurs after the age of 40, but the condition for experiment participants was defined as being over 18 years old. Is there a special reason?
|
Thanks a lot, for pointing out this issue. Although CMCJ OA is an age-related condition, previous systematic reviews and RCTs in CMCJ-OA have used in their inclusion criteria patients >18 years old. Therefore, we followed the same criterion. |
|
It is believed that comparative analysis of the effects of interventions will be possible only when the cause of OA is the same. Please also provide criteria for the cause of OA in the selection or exclusion criteria. |
The criteria of the cause of OA were provided as recommended. ‘The population included only adult patients of both sexes with thumb CMCJ OA due to degeneration, overuse or injury.’ |
|
In Comparison, there is an expression 'other type of exercise', but for the reader's clear understanding, it is necessary to know the specific meaning. |
To improve readers’ clear understanding, the paragraph rephrased as recommended.
‘We included studies that compared exercise-based interventions with any type of non-surgical treatment, control, placebo or sham-interventions. Also, we included studies including comparisons between different types of training programmes (i.e. strengthening, proprioceptive, nerve gliding, supervised or unsupervised etc.).’
|
|
We believe that accurate standards are important for selecting and extracting data in review papers. Discrepancies in data selection and extraction are described, and it is stated that a consensus process was conducted including a third reviewer. Please describe by what standards the consensus process, including a third reviewer, was conducted. |
We rephrased the data extraction and selection paragraphs to include the standards of the consensus process.
To resolve any discrepancies a consensus process was followed including the following standards: If the two reviewers have reached different judgements, the conflict was resolved by a discussion between them. If there was no consensus a referral to a third reviewer (SK) was made for the final decision. All disagreements were recorded in a separate record. |

Reviewer 2 Report
Comments and Suggestions for Authors
The current manuscript aims to assess the effectiveness of Exercise-based interventions in the management of patients with thumb carpometacarpal osteoarthritis through a systematic review and meta-analysis. The study found that exercise interventions are effective. The manuscript is well-written and informative. However, it needs extensive revisions and modifications to be publishable. Please find the following comments:
1- Title: Well written
2- Abstract:
line 27-29: ‘’ Based on very low and low certainty of evidence, proprioceptive exercises are not clinically better than standard care in man aging the condition’’. This seems a conclusion but is not related to study aim to be a conclusion.
3- Introduction:
A- What do the current evidence tell? What this review can add?
B- Line 52-54: ‘’ Our systematic review aimed to evaluate the effectiveness of exercise-based interventions compared with other non-surgical interventions in the management of patients with thumb CMCJ OA’’. This is not in line with that you described in the inclusion criteria
4- Methods:
A- Line 64-65: What is the additive intervention?
B- Line 68-73: It seems as inclusion criteria for interventional study but not review
C- Line 114-115: ‘’ Continuous data for primary and secondary findings were converted into 100-point scales’’ Why?
D- Line 115: ‘’ We used mean differences (MDs) or standardised mean differences (SMDs)’’. Justify the selection of effect size unit.
E- The use of Random-effect model is sound
F- The interpretation of MCID is not clear. What did you adapt? Effect size or upper/lower limit??. Support the interpretation with a reference
5- Results:
A- Line 169-170: not clear
B- The text did not have the meta-analyses results.
C- The results are very long and not informative. Completely revise. I suggest transforming the GRADE tables to the appendix. The text must be more informative
D- What is indirect comparison. This is not network meta-analysis.
E- I suggest add the table of the main findings (See: https://doi.org/10.1097/phm.0000000000000970)
6- Discussion:
The discussion must discuss the mechanism of the effect of the exercises. It is not clear. The discussion seems to be rereporting of the results.
7- Conclusion:
A- the conclusion about proprioceptive exercises is nor clear
Author Response
|
Comments |
Responses |
|
Reviewer The current manuscript aims to assess the effectiveness of Exercise-based interventions in the management of patients with thumb carpometacarpal osteoarthritis through a systematic review and meta-analysis. The study found that exercise interventions are effective. The manuscript is well-written and informative. However, it needs extensive revisions and modifications to be publishable. Please find the following comments: |
We would like to thank the reviewer for the valuable feedback. All comments were considered and changes were made in the revised document according to your recommendations.
|
|
1- Title: Well written |
1. - 2. Thanks a lot for pointing out this issue. The sentence was excluded from the revised manuscript.
|
|
3- Introduction:
|
3 A. We have been rephrased the paragraph as recommended. ‘Previously published systematic reviews supported that therapeutic exercises along with manual therapy and orthotic interventions can be effective on improving pain and func-tion at short-term follow-up in patients with CMCJ OA [5,6]; however, their recommenda-tions should be considered cautiously due to the low methodological quality and the lim-ited numbers of trials included. In the same line, Spaans et al. (2020) in their systematic review advocated that ‘hand therapy’ and intraarticular injections are effective interven-tions in the management of CMCJ OA at least in the short-term [7]; nevertheless, the effect of hand exercise programmes was poorly reported. Also, several trials evaluating the effectiveness of different training programmes in CMCJ OA have been published recently [8-10]; however, they have not been considered in a sys-tematic review yet. Therefore, a research synthesis on the effectiveness of exercises com-pared with other interventions seems necessary [5,8].
B. To be in line with the inclusion criteria we described that ‘Studies including patients managed surgically or diagnosed with rheumatic diseases, peripheral or central neurological deficits, De Quervain tenosynovitis or Dupuytren’s syndrome were excluded from the present review ‘
Interventions included any type of exercise, such as stretching, isometric, isotonic, concentric or eccentric contractions (or a combination thereof). Exercises could include equipment such as elastic bands, elastic bars, dumbbells or squeeze balls. The exercise programmes could be used alone or in addition to other interventions. Also, they could be supervised or prescribed as a home exercise programme. Comparisons We included studies that compared exercise-based interventions with any type of non-surgical treatment, control, placebo or sham-interventions. Also, we included studies including comparisons between different types of training programmes (i.e. strengthening, proprioceptive, nerve gliding, supervised or unsupervised etc.).
|
|
- Methods:
|
4-Methods A. A -The sentence was rephrased as suggested. ‘Trials should evaluate the effectiveness of exercises alone or as an additive intervention to a multimodal therapy programme compared with other conservative interventions.’
|
|
B- Line 68-73: It seems as inclusion criteria for interventional study but not review
|
B- The section was rephrased 2.1.2. Participants We included studies with patients >18 years of both sexes diagnosed with thumb CMCJ OA (clinically, radiologically or both). Studies with patients diagnosed with thumb CMCJ OA without pain or disability were excluded. In the same line, we excluded studies with patients treated surgically or diagnosed with rheumatoid arthritis, carpal tunnel syndrome, De Quervain’s tendonitis or Dupuytren’s contracture were excluded.
|
|
C- Line 114-115: ‘’ Continuous data for primary and secondary findings were converted into 100-point scales’’ Why? |
The sentence was rephrased as follows: ‘To facilitate data synthesis all continuous data were converted into 100-point scales.’ |
|
D- Line 115: ‘’ We used mean differences (MDs) or standardised mean differences (SMDs)’’. Justify the selection of effect size unit. |
The sentence was rephrased to justify the selection of effect size unit. ‘To evaluate the treatment effect, we used the mean differences (MDs) with 95% CIs when the outcome was measured in the same unit or the standardised mean differences (SMDs) with 95% CIs for different units [15]. ‘ |
|
E- The use of Random-effect model is sound |
It is indeed. |
|
F- The interpretation of MCID is not clear. What did you adapt? Effect size or upper/lower limit? Support the interpretation with a reference |
Thank you for pointing out this issue. We rephrased the paragraph as recommended. ‘Franchignoni et al. (2013) using the receiver-operating-characteristic (ROC) curve ap-proach evaluated the minimal clinically important differences (MCIDs) of the DASH and QuickDASH that were set at a mean 11- and 16-points change, respectively [18]. Using a similar approach, Sallafi et al. (2004) reported the MCID for pain intensity in patients with hand osteoarthritis at 15% improvement from the pooled weighted mean of the baseline[19]. Based on another study, the MCID of the grip and pinch strength in patients with hand OA were reported at 0.84 kg and 0.33 kg with large effect sizes (Cohen’s d>1), respectively [20].’ |
|
5- Results:
|
A- The sentence was rephrased as suggested. ‘Nine of the eligible trials evaluated the use of proprioceptive exercise programmes [21-29].’
B- We included in the text the meta-analysis results as recommended.
C- We have revised the text to be more informative. We moved the GRADE tables in the supplementary material as requested.
D- The sentence was rephrased as recommended ‘Nine RCTs [22-27,29,32,33] compared proprioceptive exercises with standard therapy for thumb CMCJ OA’
|
|
E- I suggest add the table of the main findings (See: https://doi.org/10.1097/phm.0000000000000970) |
Possibly, the reviewers did not have access to the table of the main findings because it has been included in a supplementary material. To improve readers’ clarity, we included it as a Table in the revised manuscript as recommended. |
|
6- Discussion: |
Thank you for the suggestions. We included a discussion about the mechanisms of the effect of the exercises.
‘It is well documented that different types of exercises, such as aerobic, neuromuscu-lar, proprioceptive and resistance training, can act as pain-modulation interventions in patients with OA [46]. The most significant underlying mechanism for this phenomenon is considered the activation of descending inhibitory pathways resulting in widespread hypoalgesia following even after a single bout of exercise [33,46]. Considering that signif-icant widespread sensitization has been identified in patients with CMCJ OA, exercise induced hypoalgesia can be a valuable treatment option to improve the treatment out-comes in patients with the condition [23]. Also, exercise-based interventions have been reported to relieve several pathological mechanisms of OA [46]. Some of these mecha-nisms include the degradation of the extracellular matrix, apoptosis, inflammation and other cellular changes [46-49].
‘The presence of pain in patients with thumb CMCJ OA significantly affects functional activities and is usually the most important reason for patients seeking rehabilitative ser-vices [6,40]. Therefore, interventions including manual therapy or orthotic interventions are commonly used to decrease pain in the current patient group [24,39]. However, longstanding OA involves complex pathophysiological mechanisms resulting in altered pain transmission and several changes within the joint afferent neurons and the central nervous system [50-52]. Thus, there is increased recognition that the management of OA interventions should include exercises and education regarding pain to effectively de-crease pain intensity, increase self-efficacy and improve social functioning [50]. Contem-porary evidence suggests that the presence of osteoarthritis is associated with a proprio-ceptive deficit in thumb CMCJ OA [32,53]. A logical assumption is that proprioceptive ex-ercises may enhance joint stability, improve osteoarticular coordination, increase func-tional performance in daily activities and result in pain reduction. Nevertheless, based on the present findings, the use of proprioceptive exercises does not present superior clinical benefits compared to standard care in thumb CMCJ OA.’ |
|
7- Conclusion: |
Conclusion was rephrased to become clear regarding the use of proprioceptive exercises. ‘Based on the available data, exercise-based interventions should be used as a first-line treatment in patients with thumb CMCJ OA providing statistically and clinically significant benefits compared with no treatment, at least in the short term. Also, the use of proprioceptive exercises does not seem to provide clinically better outcomes compared to other types of exercises. Further research evaluating the effect of exercises including different patient subgroups seems necessary.’
|

Reviewer 3 Report
Comments and Suggestions for Authors
Manuscript ID: healthcare-2914866
Manuscript title: Exercise-based interventions are effective in the management of patients with thumb carpometacarpal osteoarthritis: a systematic review and meta-analysis of randomised controlled trials
Comments
This manuscript reports a systematic review with meta-analysis aimed to evaluate the effectiveness of exercise-based interventions compared with other non-surgical interventions in the management of patients with thumb carpometacarpal joint osteoarthritis (CMCJ OA). Subgroup analyses was performed to compare exercises with different types of intervention, control or other types of exercises at different follow-up . The reporting followed appropriate guidelines and reported all relevant information. The trial was registered as PROSPERO (current status: ongoing). I have a few comments for the authors to consider.
Major comments
1. lines 121-122, Section 2.6 Data Analysis, synthesis and summary findings. Please rephrase this sentence. I-sqr assesses heterogeneity – non-random variation – in treatment effect among studies included in a meta-analysis.
2. Figure 1. The number of records excluded should read 15 (not 25).
3. The manuscript showed a 13% similarity with a published paper (https://doi.org/10.7759/cureus.48907) authored by 2 (out of 4) of the same authors. Please double-check to minimize the similarity.
Author Response
We would like to thank the reviewer for the comments. All comments were considered and changes were made in the revised document according to your recommendations.
|
Comments |
Responses |
|
Reviewer This manuscript reports a systematic review with meta-analysis aimed to evaluate the effectiveness of exercise-based interventions compared with other non-surgical interventions in the management of patients with thumb carpometacarpal joint osteoarthritis (CMCJ OA). Subgroup analyses was performed to compare exercises with different types of intervention, control or other types of exercises at different follow-up . The reporting followed appropriate guidelines and reported all relevant information. The trial was registered as PROSPERO (current status: ongoing). I have a few comments for the authors to consider.
|
Thank you for your valuable feedback.
|
|
Comments
Major comments
1. lines 121-122, Section 2.6 Data Analysis, synthesis and summary findings. Please rephrase this sentence. I-sqr assesses heterogeneity – non-random variation – in treatment effect among studies included in a meta-analysis.
2. Figure 1. The number of records excluded should read 15 (not 25).
3. The manuscript showed a 13% similarity with a published paper (https://doi.org/10.7759/cureus.48907) authored by 2 (out of 4) of the same authors. Please double-check to minimize the similarity.
|
1. Thanks a lot for the suggestion. The phrase was changed according to your recommendations as follows: ‘The I2 statistic that aims to indicate the proportion of the non-random variation in the treatment effect was used to estimate the heterogeneity among the studies included in the meta-analysis [9]. I2 values ≥0.75 were considered of high heterogeneity [9].’
2. We are very sorry for the typo. It was corrected.
3. Similarity was minimized as suggested. |
|
|
|
|
|
|
|
|
|

Round 2
Reviewer 2 Report
Comments and Suggestions for Authors
The comments have been well addressed. My only comments are:
1- The conclusion in current form is totally unaccepted. The results can not recommend or conclude what is the first line treatment.
2- Line 235: Correct the 95% CI. The upper or lower limit is missing
3- The interpretation of MCID is still not clear. What did you adapt? Effect size or upper/lower limit? Support the interpretation with a reference.
Less...
Author Response
|
Comments |
Responses |
|
The comments have been well addressed. My only comments are: |
Thanks a lot for the second round of the review. We have considered all comments and revised the manuscript as recommended. |
|
1- The conclusion in current form is totally unaccepted. The results can not recommend or conclude what is the first line treatment. |
We agree with your comment and rephrased the conclusion as follows: ‘Based on the available data, exercise-based interventions provide statistically and clinically significant benefits in pain reduction and wrist disability compared with no treatment in patients with thumb CMCJ OA in the short term. Also, the use of proprioceptive exercises provides statistically and clinically better outcomes compared to other types of exercises only in pain intensity at the short-term without further differences in the outcomes at mid- and long-term follow-up times. Future research evaluating the effect of exercise-based interventions in different patient subgroups is necessary.’ |
|
2- Line 235: Correct the 95% CI. The upper or lower limit is missing |
The sentence was rephrased as recommended,
One RCT [32] compared a radial nerve mobilisation exercise and a placebo intervention, suggesting significant differences in pressure pain thresholds at the CMCJ in favour of the experimental group at very short-term follow-up (MD: 1.58, 95%CI: 0.73, 2.42; p<0.001). |
|
3- The interpretation of MCID is still not clear. What did you adapt? Effect size or upper/lower limit? Support the interpretation with a reference. |
Thanks a lot for the comment. We are very sorry for the confusion regarding MCIDs calculation. We used effect sizes to calculate MCIDs with Cohen d>0.5. We included in the results the effect size calculation and rephrased where necessary the clinically significant results.
‘To evaluate the minimal clinically important differences (MCIDs) we have used the effect size value [17]. Based on Cohen’s d benchmark, we considered effect sizes below than 0.2 as low, 0.5 as medium and 0.8 or higher as high [18]. We calculated MCIDs by dividing the mean difference in scores by the standard deviation of baseline scores and values of 0.5 were considered as clinically significant [19]. |

Reviewer 3 Report
Comments and Suggestions for Authors
Thank you for the opportunity to review your masnucript. All comments were adequately addressed. I have no new comments.
Author Response
We are grateful for your help.